# Robust and memory-less median estimation for real-time spike detection

**Ariel Burman**[1], **Jordi Solé-Casals**[2,3], **Sergio E. Lew**[1] *

**1** Instituto de Ingeniería Biomédica, Universidad de Buenos Aires, Buenos Aries, Argentina, **2** Data and Signal Processing Group, University of Vic-Central University of Catalonia, Vic, Spain, **3** Department of Psychiatry, University of Cambridge, Cambridge, United Kingdom

* slew@fi.uba.ar

**Data Availability Statement:** All relevant data for this study are publicly available from the Instituto de Ingeniería Biomédica in the Github repository (https://github.com/IIBM/fast-median).

## Abstract

We propose a novel 1-D median estimator specifically designed for the online detection of threshold-crossing signals, such as spikes in extracellular neural recordings. Compared to state-of-the-art algorithms, our method reduces estimator variance by up to eight times for a given buffer length. Likewise, for a given estimator variance, it requires a buffer length that is up to eight times smaller. This results in three significant advantages: the footprint area decreases by more than eight times, leading to reduced power consumption and a faster response to non-stationary signals.

## Introduction

Median filtering is a widely used technique for removing or detecting infrequent events in signal and image processing. Depending on which part of the raw signal is removed, the median filter can be utilized to either eliminate or capture impulse noise from signals and images [1, 2], such as in the case of extracellular neuronal spikes or "pepper and salt" noise, respectively. When sparse events are contaminated with Gaussian noise, as is the case with extracellular neuron recordings, an optimal threshold can be calculated to distinguish spikes from noise [3, 4]. Indeed, most state-of-the-art neuronal decoding techniques begin with threshold-based spike detection, regardless of the type of analysis performed afterward, such as multi-unit or single-unit spike sorting and isolation decoding [4–12].

While computing the median-based threshold in offline spike detection analysis is straightforward, the only available solution for real-time neuronal decoding is a sliding (or moving) median estimator. An efficient moving median estimator maintains both an ordered buffer of the last $L$ samples and an index buffer that records the arrival time of each sample. In a sequential implementation, this solution requires $\mathcal{O}(\log L)$ steps to determine the position of the new sample within the ordered buffer. Alternately, parallel implementations for comparing the new sample with the entire buffer reduce complexity to $\mathcal{O}(1)$ [13–17]. However, both implementations rely on auxiliary information to track the oldest element in the buffer.

Here, we present an alternative method that not only eliminates the need for this auxiliary information buffer but, more importantly, requires a significantly smaller buffer to achieve a given level of accuracy. This approach drastically reduces both hardware area-size and power

**Funding:** This study was financially supported by Secretaria de Ciencia y Tecnica, Universidad de Buenos Aires [https://cyt.rec.uba.ar/] in the form of a grant (20020220200032BA) received by SEL. No additional external funding was received for this study. The funder had no role in study design, data collection and analysis, decision to publish, or preparation of the manuscript.

**Competing interests:** The authors have declared that no competing interests exist.

consumption. We demonstrate that the proposed algorithm is unbiased and more robust than the traditional moving median algorithm. Furthermore, for substantial changes in the input signal, it adapts to the new probability distribution significantly faster than traditional methods, primarily due to its shorter buffer length.

This paper is organized as follows: In the Materials and methods section, we describe the algorithm, the insertion and dropping rules, and provide examples of its behavior. Additionally, we present a hardware implementation that demonstrates the improvement in area-size. In the Results section, we compute the estimator variance and quantify its performance gains compared to the classical moving median across various probability distributions.

## Materials and methods

Let $X(t)$ be a continuous random process, with probability distribution density function $f_X(x)$, which we will only assume it is uni-modal. The classical moving median estimator can be computed, at time $t_n$ as

$$\mu_{t_n} = median\{x_{t_n}, x_{t_{n-1}}, \ldots, x_{t_{n-(L-1)}}\} \tag{1}$$

when using a buffer of length $L$.

Given a buffer $B$ with $L$ positions, being $L = 2m + 1$ so $x_{m+1}$ is the element in the middle. For every pair $(i, j)$ given that $i < j$, $x_i \leq x_j$ at all time. In Fig 1a-upper the ordered buffer is displayed with elements from 1 to $2m + 1$ at time $k$.

When a new sample, denoted by $x_{new}^{k+1}$, arrives at time step $k + 1$ and is lower than the median $x_{m+1}^k$, in the buffer, one of two scenarios occurs:

- Insertion within the buffer: if $x_{new}^{k+1}$ is equal to or greater than at least one element, $x_i^k$ (where $1 \leq i \leq m$), in the buffer, it is inserted at the appropriate position to maintain the ascending order of the buffer.

- Insertion at the beginning: if $x_{new}^{k+1}$ is lower than all elements in the buffer, it becomes the new minimum and is inserted at the beginning of the buffer.

In both cases, the current maximum element, $x_{2m+1}^k$, is removed from the buffer to maintain its fixed size, as shown in Fig 1a-middle. The opposite case is shown in Fig 1a-lower. Finally, when the new sample is equal to the element in the center of the buffer, either the first or the last element of the buffer is randomly dropped, and the new sample is inserted in the middle.

It is important to note that in this algorithm: a) the elements in the buffer $B$ are consistently sorted, b) the new sample $x$ is always inserted into the buffer, and c) there is no requirement for information regarding the time of sample arrival.

To determine the position where the new element should be inserted, we must either locate an element of the same value in the buffer or conduct at least $\mathcal{O}(\log L)$ comparisons using a binary search algorithm. Performing a parallel comparison of the new element with all elements in the buffer simultaneously achieves $\mathcal{O}(1)$ complexity.

Fig 1b illustrates an example of insertions into a sorted buffer using both algorithms: our newly proposed median (NM) in the left column and the classical moving median (CMM) in the right column. In the CMM example, the first row depicts the buffer, while the second row indicates the lifetime of each element.

The sample column indicates which element has arrived and been inserted at each time step. The first sample, 2, is lower than the central element, so it must be inserted in an orderly fashion on the left side of the buffer, while the last element, 7, is dropped. The central element is now 3. The NM highlights in green where the new sample was inserted and in red the

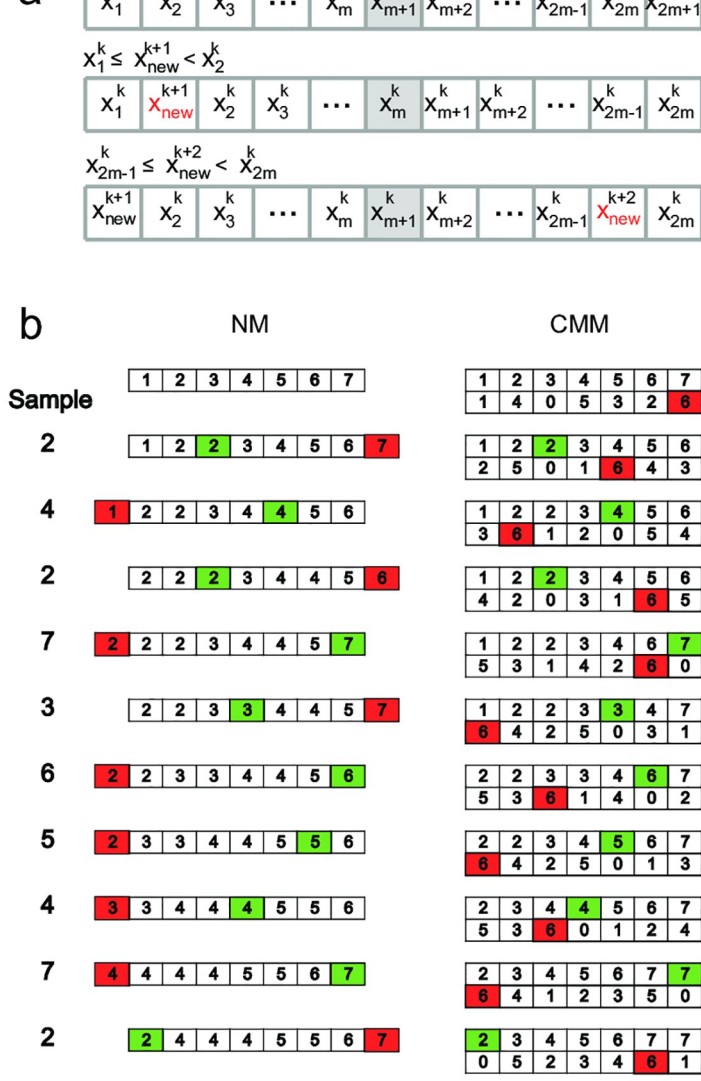

**Fig 1. Median estimates of an extracellular recording.** (a) The algorithm is illustrated during three time steps. (b) Example evolution of the buffer with a stream of samples.

element that has been dropped. In the next step, the sample 4 arrives, which is higher than the median 3. Therefore, it must be inserted on the right side of the buffer, pushing all elements lower than 4 to the left and removing the first element, which is 1. As a result, 4 occupies the central position.

For the CMM algorithm, we begin with the buffer in the same initial state as with the NM (first row). Each sample is assigned its corresponding lifetime in the buffer (second row). Before a new sample arrives, we already know which element will be dropped. In the CMM algorithm, the red tag indicates the oldest element, which will be removed prior to the insertion of the new sample. The new sample is highlighted in green to indicate its position in the buffer. In the first step, the oldest element, 7, is removed. The lifetimes of all remaining elements are increased by 1. A new element is then inserted into the buffer in the appropriate

order, with an arrival time of 0. In the next step, the oldest element in the buffer is 4, which will be removed to make room for the new sample, 2, to be inserted in the correct order.

After several iterations, the NM buffer is primarily populated with values close to the median, while the CMM buffer retains the last N elements drawn from the original distribution.

We have uploaded a freely available software implementation of the new algorithm, along with examples that illustrate the dynamics of the buffer (http://github.com/IIBM/fast-median).

## Hardware design

In Fig 2 we present the logic-level design of our median estimator. Each cell of a *L*-length buffer a *n*-bit register ($R_i$) where the data is stored, a simple *n*-bit comparator defined as $C_i$: $R_i \leq X => 0; R_i > X => 1$, a multiplexer with inputs $X$, $R_i$, $R_{i-1}$ and $R_{i+1}$ to select which element is placed in each position of the buffer with every new sample, and a small combinational circuit to control the multiplexer. The central cell, where the median *M* resides, has a full *n*-bit comparator (as opposed to a simpler version), which generates three independent output signals, each corresponding to the conditions $X < M$, $X = M$, $X > M$. Buffer control depends on these three signals, which are mutually exclusive.

Each section of the buffer -left side, median, and right side- is illustrated in Fig 2 with its corresponding truth table from the combinational circuit that controls the multiplexer. Increasing the buffer length requires adding an equal number of cells to each side of the median, along with the correspondent configuration. Since the signals from the full comparator are mutually exclusive, the invalid cases in the truth tables are shaded in gray, and the

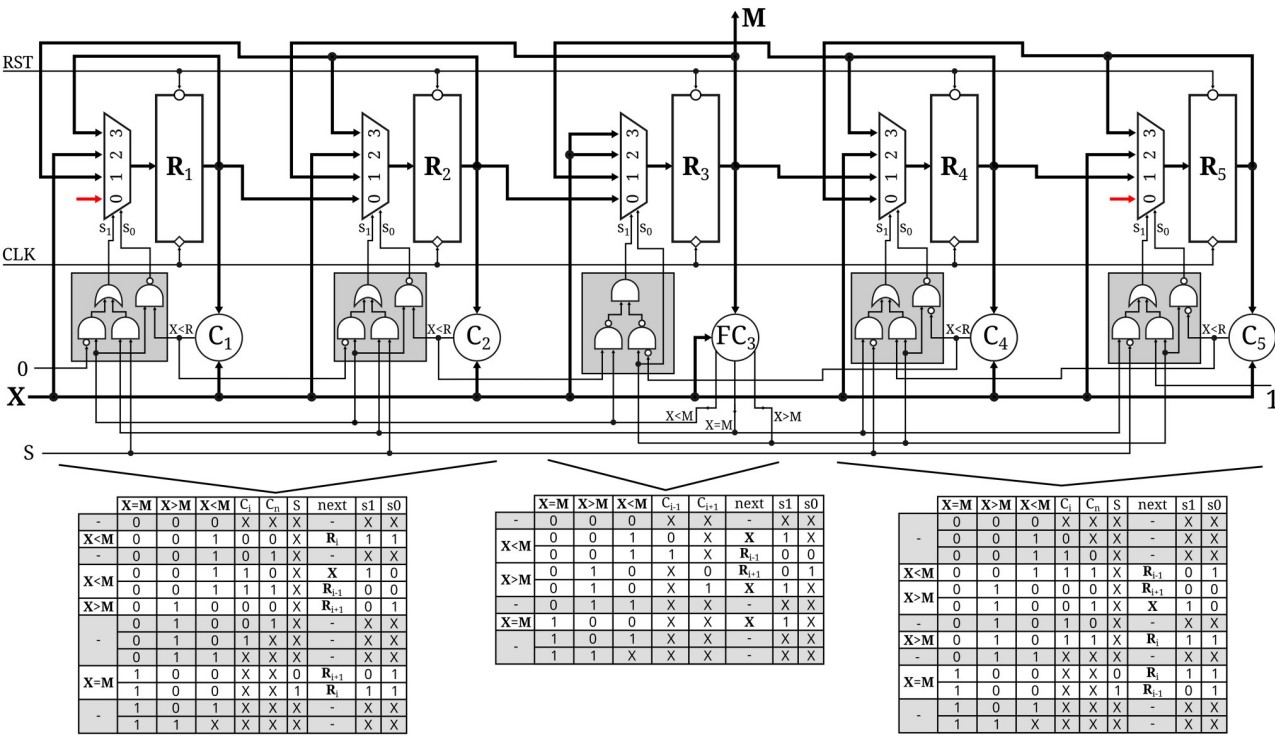

**Fig 2. Hardware implementation.** The buffer is divided in three section: left(lower)-side, middle(median) and right(high)-side. Cells from same section have same combinational logic circuit which controls the multiplexer.

| X=M | X>M | X<M | $C_i$ | $C_n$ | S | next | s1 | s0 |
|---|---|---|---|---|---|---|---|---|
| - | 0 | 0 | 0 | X | X | X | - | X | X |

| | X=M | X>M | X<M | $C_i$ | $C_n$ | S | next | s1 | s0 |
|---|---|---|---|---|---|---|---|---|---|
| - | 0 | 0 | 0 | X | X | X | - | X | X |
| X<M | 0 | 0 | 1 | 0 | 0 | X | $R_i$ | 1 | 1 |
| - | 0 | 0 | 1 | 0 | 1 | X | - | X | X |
| - | 0 | 0 | 1 | 1 | 0 | X | $X$ | 1 | 0 |
| X<M | 0 | 0 | 1 | 1 | 1 | X | $R_{i-1}$ | 0 | 0 |
| X>M | 0 | 1 | 0 | 0 | 0 | X | $R_{i+1}$ | 0 | 1 |
| - | 0 | 1 | 0 | 0 | 1 | X | - | X | X |
| - | 0 | 1 | 0 | 1 | X | X | - | X | X |
| - | 0 | 1 | 1 | X | X | X | - | X | X |
| X=M | 1 | 0 | 0 | X | X | 0 | $R_{i+1}$ | 0 | 1 |
| X=M | 1 | 0 | 0 | X | X | 1 | $R_i$ | 1 | 1 |
| - | 1 | 0 | 1 | X | X | X | - | X | X |
| - | 1 | 1 | X | X | X | X | - | X | X |

| | X=M | X>M | X<M | $C_{i-1}$ | $C_{i+1}$ | next | s1 | s0 |
|---|---|---|---|---|---|---|---|---|
| - | 0 | 0 | 0 | X | X | - | X | X |
| X<M | 0 | 0 | 1 | 0 | X | $X$ | 1 | X |
| | 0 | 0 | 1 | 1 | X | $R_{i-1}$ | 0 | 0 |
| X>M | 0 | 1 | 0 | X | 0 | $R_{i+1}$ | 0 | 1 |
| | 0 | 1 | 0 | X | 1 | $X$ | 1 | X |
| - | 0 | 1 | 1 | X | X | - | X | X |
| X=M | 1 | 0 | 0 | X | X | $X$ | 1 | X |
| - | 1 | 0 | 1 | X | X | - | X | X |
| - | 1 | 1 | X | X | X | - | X | X |

| | X=M | X>M | X<M | $C_i$ | $C_n$ | S | next | s1 | s0 |
|---|---|---|---|---|---|---|---|---|---|
| - | 0 | 0 | 0 | X | X | X | - | X | X |
| | 0 | 0 | 1 | 0 | X | X | - | X | X |
| | 0 | 0 | 1 | 1 | 0 | X | - | X | X |
| X<M | 0 | 0 | 1 | 1 | 1 | X | $R_{i-1}$ | 0 | 1 |
| X>M | 0 | 1 | 0 | 0 | 0 | X | $R_{i+1}$ | 0 | 0 |
| | 0 | 1 | 0 | 1 | 0 | X | - | X | X |
| X>M | 0 | 1 | 0 | 1 | 1 | X | $R_i$ | 1 | 1 |
| - | 0 | 1 | 1 | X | X | X | - | X | X |
| X=M | 1 | 0 | 0 | X | X | 0 | $R_i$ | 1 | 1 |
| X=M | 1 | 0 | 0 | X | X | 1 | $R_{i-1}$ | 0 | 1 |
| - | 1 | 0 | 1 | X | X | X | - | X | X |
| - | 1 | 1 | X | X | X | X | - | X | X |

output of the combinational circuit is marked as X. Each cell receives the output from its self-comparator, $C_i$, and the comparator of the adjacent cell, $C_n$. For the left side of the buffer $C_n$ refers to $C_{i-1}$ while for the right side, it refers to $C_{i+1}$. The central cell receives inputs from both the left and right cell comparators.

As we explained when describing the algorithm, when a new sample arrives, assuming that $X < M$, the new value should be inserted on the left side of the buffer. From the position where it is inserted, all subsequent values will be shifted to the right, removing the last cell's value. This means that all the cells on the right side of the buffer will shift their values, as indicated in the left-side truth table, which represents the only possible action. On the left side, each cell compares the value $X$ to its register $R_i$ and generates $C_i$. For a given cell, if $C_i = 0$, it indicates that $X$ is equal to or greater than the cell's register $R_i$. Since we are in the lower section of the buffer, the new value should be inserted to the right of this cell, and this position should remain unchanged. Conversely, if $C_i = 1$, it means that the cell's value is greater than the new sample. In this case, either this cell will store the new value, or the new value will be stored in a lower cell, in which case this cell should take the value of $R_{i-1}$. Both scenarios are defined by the comparator in the next lower cell, using $C_n = C_{i-1}$. The first element of the buffer always receives $C_n = 0$. This will result in the new element being inserted in the first position if $C_1 = 1$ or retaining the same value if $C_1 = 0$. For this reason, the multiplexer input for $R_{i-1}$ is not used. Analogously, behavior explains the truth table for the right side of the buffer.

The central cell follows similar behavior when using either $C_{i-1}$ or $C_{i+1}$, depending on whether $X$ is lower or higher than the current value. For the special case when $X = M$, the new value is always inserted at the central position, and the $S$ signal is used to determine whether to drop the first or the last value, thereby shifting either the left or the right side of the buffer, respectively. This signal can take on a value of either 1 or 0. Several strategies can be employed to generate $S$ but a single alternating bit is sufficient to ensure the unbiasedness of the estimator.

## Results

While median filtering finds applications in various domains, it is especially valuable for detecting extracellular neuron spikes. In an extracellular recording setup, electrodes capture the combined electrical activity of numerous distant neurons. Regardless of the individual probability distributions of these discrete sources, the central limit theorem dictates that their collective contribution approximates a normal distribution. Consequently, the stronger the electrical field perturbation caused by a nearby neuronal spike on an electrode, the greater the likelihood of identifying it as a single-cell action potential. Thus, it is customary to define the signal-to-noise ratio (SNR) between spikes and background noise as the ratio of the peak voltage of the spikes to the standard deviation of the background noise, denoted as $\sigma_{noise}$. A SNR value of K indicates that the peak voltage of the spike is K times the standard deviation of the background noise, denoted as $\sigma_{noise}$. In the case of Gaussian noise $N(0, \sigma_{noise}^2)$, a linear relationship exists between its standard deviation and the median of the half-normal distribution, represented as $y = |x|$,

$$\sigma_{noise} = \frac{\text{median}\{|x|\}}{\sqrt{2} \; erf^{-1}(1/2)} \approx \frac{\text{median}\{|x|\}}{0.675} \tag{2}$$

Thus, calculating a threshold $T$ to detect events surpassing $T$ reduces the computation of $\sigma_{noise}$ to the median of $y = |x|$, where $p_X(x) = N(0, \sigma_{noise}^2)$. In Fig 3, we compare the classical moving median estimator (red) with our proposed method (green), which exhibits a lower variance when applied to real neuronal extracellular recordings. This reduction in variance results

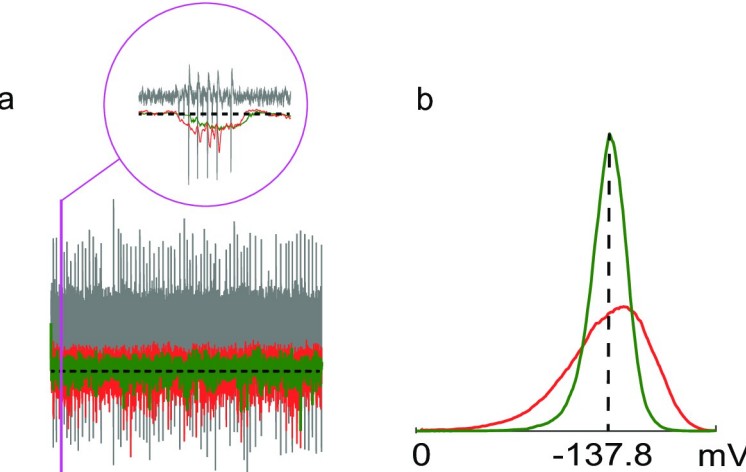

**Fig 3. Detecting neural spikes by thresholding.** (a) The figure shows a segment of a real extracellular recording from the prefrontal cortex of a rat [18]. The recording is displayed in gray. Thresholds estimated using the classical moving median estimator and the proposed estimator are shown in red and green, respectively. The dashed line represents the threshold computed from the entire recording, set at 4 $\sigma_{noise}$. A zoomed-in view of the area within the purple rectangle is provided to highlight the neural spikes. (b) The value distribution for both the classical and the new median estimators is illustrated. As observed, the classical median estimator exhibits a longer tail towards negative values due to its sensitivity to outliers caused by spikes.

in fewer missed spikes and eliminates the erroneous detection of non-existent spikes, which is particularly crucial for accurate spike detection through thresholding.

The reduction in variance in the estimation is attributed to the fact that samples are discarded from the buffer based on their position within it, rather than their arrival times. Specifically, in an initially ordered buffer populated with the first $L$ samples, the classical moving median (CMM) estimator discards the oldest sample in the buffer each time a new sample arrives. In contrast, our algorithm (NM) discards a sample from the extremes of the buffer. Intuitively, this can be understood as a process of sample selection that compresses the data, ultimately resulting in a buffer filled with samples that are closer to the median.

It is important to note that the classical algorithm preserves the true sampling distribution of the original signal, whereas our method does not completely represent this distribution. We reasoned that while this characteristic may provide an advantage in a stationary process, it could lead to a delay in reaching the new median value when a change in the median signal occurs. Furthermore, the delay in accurately estimating the median and the length of the buffer are interconnected. Larger buffers do indeed result in less dispersion around the true median, offering greater stability, albeit at the expense of slower responses to process changes. To investigate this phenomenon, we conducted a series of simulations using different buffer lengths. In Fig 4a we generated data from a $N(\mu = 8, \sigma = 2)$ and at $time = 0$, there is a subtle change to $N(\mu = 10, \sigma = 2)$. As the normal distribution has its mean equal to its median, both estimators must converge to $\mu = 10$. Here, we used the NM with $L = 63$, and CMM for 63, 511, 1023. After sufficient time, both algorithms reach $\mu = 10$ (see S1 Fig for a similar situation with non-symmetrical distributions).

In Fig 4b we repeat this simulation 100,000 times for NM and CMM with buffer sizes of 63, 127, 255, 511, and 1023. For each iteration, we record the last value of the estimator. The estimation error, in this case, is defined as the difference between the last value of the estimator and the expected median, and we plot the distribution of this estimation error. We observe

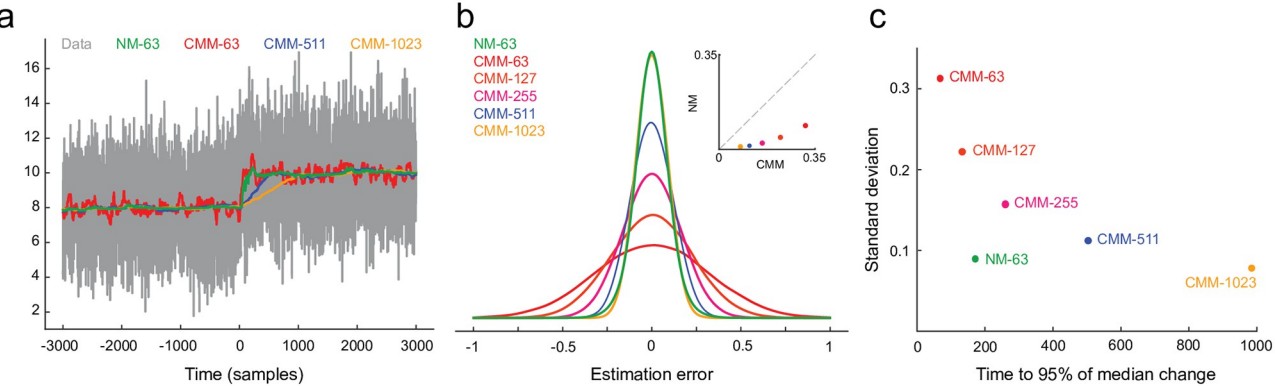

**Fig 4. Time and spatial sensitivity to process changes.** (a) The gray trace represents a normal distribution, denoted as $N(\mu = 8, \sigma = 2)$, with a median change occurring at $t = 0$. The changes in the median estimator for buffers of lengths 63, 511, and 1023 are plotted in green, red, blue, and yellow, respectively. (b) Dispersion around the true median value for different buffer lengths, including our algorithm (NM-63). (c) The compromise between dispersion and the time required to reach 95% of the new median value following a change.

that NM-63 exhibits a distribution similar to that of CMM-1023. In the upper-right panel of Fig 4b we matched the standard deviations of these distributions and paired same-length buffers for each algorithm, demonstrating that NM consistently outperforms CMM in terms of error deviation for buffers of the same length.

In Fig 4c we analyzed the time (measured in the number of samples) it takes for each algorithm to reach the new median after a step change. Intuitively, the CMM will achieve a new median estimation in $L$ samples. As it discards values, samples from the new distribution begin to fill the buffer. Once $L$ new samples have arrived, the buffer is completely filled with samples from the new distribution. To quantify this property, we defined the settling time as the time when the output of the algorithm reaches $\mu_f - 0.05 \cdot (\mu_f - \mu_i)$. In other words, this occurs when the estimator reaches 95% of the median change. As the NM-63 outperformed the CMM-63 through CMM-1023, we aim to analyze where this estimator stands in relation to all others. In Fig 4c we observe that the NM-63 has approximately twice the settling time of the CMM-63, but it exhibits a standard deviation that is three times lower. When compared to CMMs with similar standard deviations, such as CMM-511 and CMM-1023, the settling time of the NM-63 is 2.5 to 5 times lower.

In the S3 Fig we explored the same analysis comparing NM-63 and CMM-511 for several amplitude step changes. The results indicate that while CMM-511 exhibits a consistent response across different amplitudes, the settling time of NM-63 improves with larger changes. Unlike the CMM estimator, which reflects changes in estimation within a fixed time window $L$, our algorithm's response time depends on the magnitude of the change. For significant changes involving entirely new samples, our algorithm typically requires between $L$ and $2L$ time steps to converge to the new median. Conversely, for small changes, the new samples are added to the edges of the buffer, which may result in a longer duration -exceeding $L$ steps- before they fully influence the estimate due to the discarding process.

Slow and gradual changes in data dynamics may require additional time to converge to the true median value. While this slower response to changes acts as the counterpart for robustness, the method compensates for this drawback by providing improved resolution, as illustrated in Fig 5.

To demonstrate the algorithm's robustness against outliers, we analyzed the dynamics of sample density within the buffer over time. To directly compare the NM algorithm with the

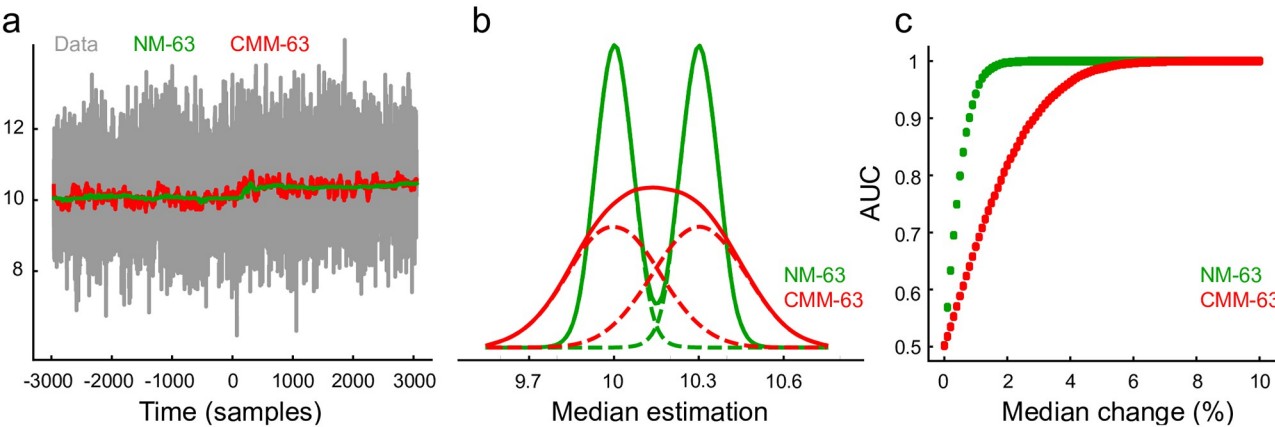

**Fig 5. Sensitivity to small changes.** (a) Online median estimation for a 3% change in the signal median. (b) (b) Dashed lines illustrate the distributions before and after the small change in the signal median. Solid lines represent the distributions of the CMM and NM estimators, respectively. As observed, while the samples of the CMM estimator exhibit an unimodal distribution, the NM estimator effectively discriminates the change. (c) ROC analysis for changes in the median process ranging from 0% to 100%.

CMM algorithm, we initially populated both buffers with $L$ pre-sorted samples. This approach ensures that the initial distribution of the NM estimator accurately reflects the underlying data. However, unlike the CMM, where this distribution remains constant, the NM estimator continuously compresses the buffer's sample distribution towards the median value, typically within a few buffer cycles.

We executed the algorithm with a buffer length of L = 1023 samples, feeding it with data drawn from a normal distribution $N(0, 1)$. Subsequently, we analyzed the sample density within the buffer at several time intervals, beginning from the moment the buffer was initially filled. Fig 6a illustrates these changes in sample density. Notably, the initially Gaussian distribution becomes increasingly concentrated around the data's mean (or median, in the case of a normal distribution), ultimately approaching a uniform distribution with significantly reduced variance.

We repeated the analysis for a folded normal distribution, which represents the distribution of $|x|$ when $x \sim N(0, 1)$. Fig 6b shows the sample distribution within the buffer over time, beginning with a buffer completely filled with fresh samples drawn from the folded normal distribution and continuing for up to 50,000 time steps. This scenario, involving a folded normal distribution, is more relevant to real extracellular recordings, where the median and mean typically differ. As can be readily observed, the buffer samples are distributed around the noise median in this case.

Interestingly, repeating both experiments 10,000 times and averaging their sample distributions across different time points reveals a consistent trend. The average sample distribution concentrates around the median value of the initial distribution, essentially copying its shape. This behavior is evident in Fig 6c for the normal distribution and Fig 6d for the folded normal distribution. Consequently, filling the buffer with values increasingly closer to the true median with each iteration enhances the method's robustness against outlier perturbations. As Fig 6 illustrates, the buffer prioritizes samples closer to the median value as time goes. This contrasts with the classical moving median estimator, where an outlier resides in the buffer for $L$ steps. In our method, outliers are relegated to the buffer's extremes and have a significantly lower probability of remaining influential for many steps, effectively halving with each step k,

$$p = \left(\tfrac{1}{2}\right)^k.$$

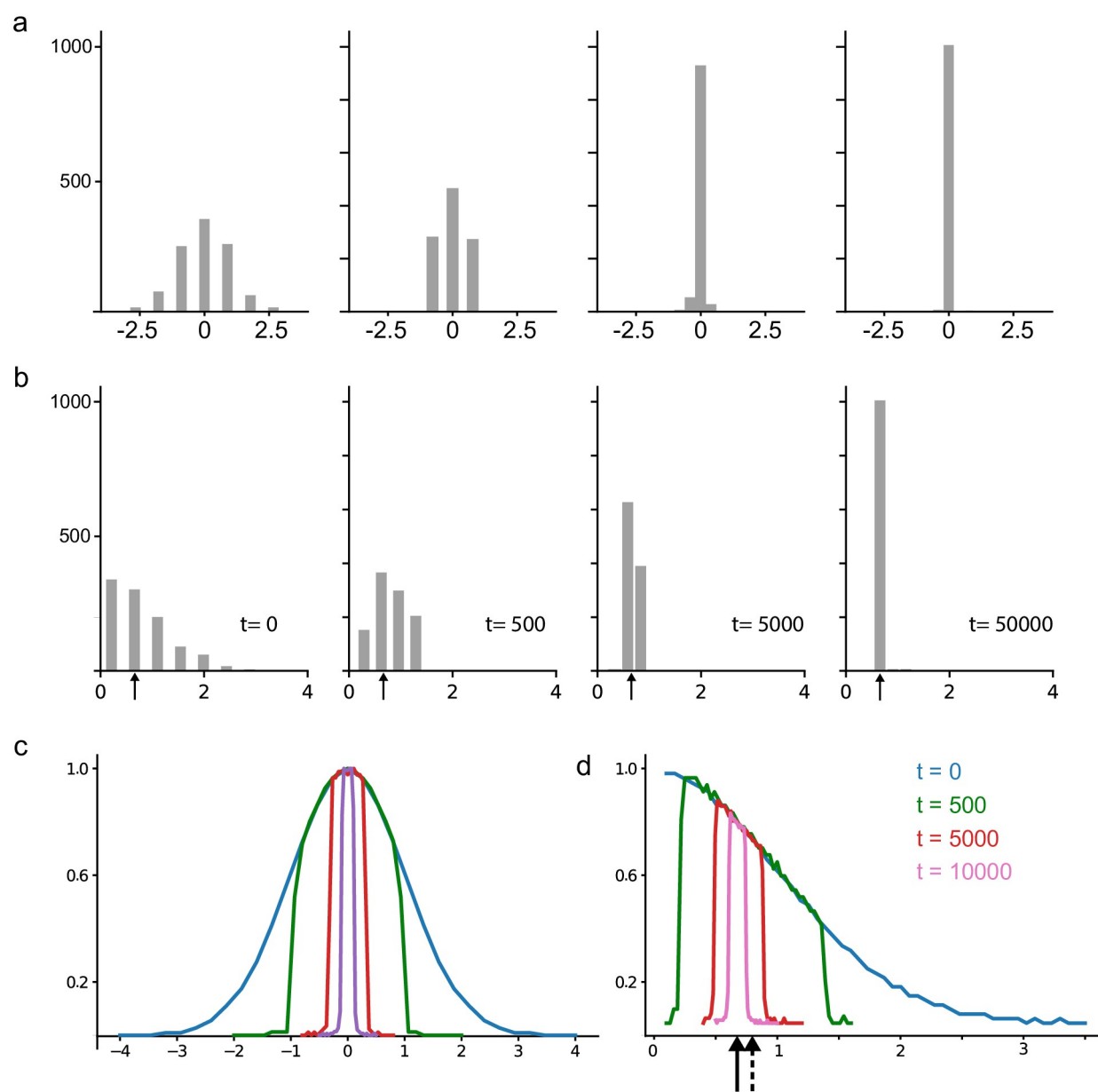

**Fig 6. Dynamical changes in the buffer population ($L$ = 1023).** (a) Sample distribution over time for $x \sim N(0, 1)$. (b) Sample distribution over time for $|x|$, when $x \sim N(0, 1)$, the arrow shows the true median of the signal. (c), (d) The sample distribution of the data across the buffer and over time is visualized through an average of 10,000 repetitions of the experiment for $x$ and $|x|$, where $x \sim N(0, 1)$. The results were normalized to highlight the magnification effect around the true signal's median value as time progresses. The continuous arrow indicates the true signal median, while the dashed arrow represents the true signal mean.

Interestingly, repeating both experiments 10,000 times and averaging their sample distributions across different time points reveals a consistent trend. The average sample distribution concentrates around the median value of the initial distribution, effectively replicating its shape. This behavior is illustrated in Fig 6c for the normal distribution and Fig 6d for the folded normal distribution. Consequently, populating the buffer with values that increasingly approach the true median with each iteration enhances the method's robustness against outlier perturbations (see S2 Fig).

As illustrated in Fig 6, the buffer prioritizes samples that are closer to the median value as time progresses. This approach contrasts with the classical moving median estimator, where an outlier remains in the buffer for $L$ steps. In our method, outliers are relegated to the extremes of the buffer, significantly reducing their probability of remaining influential over multiple steps, effectively halving with each step $k$, represented as $p = \left(\frac{1}{2}\right)^k$.

By definition, the median $m$ is the value that satisfy

$$\int_{-\infty}^{m} p(x)dx = \int_{m}^{-\infty} p(x)dx = \frac{1}{2} \tag{3}$$

for any other value $m_x$ different to $m$ we will have

$$A = \int_{-\infty}^{m_x} p(x)dx \quad ; \quad B = \int_{m_x}^{-\infty} p(x)dx \tag{4}$$

with $A \neq B$.

If the value at the center of the buffer, denoted as $m_x$, is less than the true median $m$, an imbalance will occur. The distribution function $F(x)$ indicates that there are more samples greater than $m_x$ than expected, as shown by the inequality $F(m) - F(m_x) > 0$. These additional high-value samples on the right side of the buffer tend to push $m_x$ towards the left. Consequently, this creates an opportunity for values greater than $m_x$ to occupy the central position of the buffer. Conversely, the opposite effect occurs when $m_x$ is greater than the true median.

This process can be likened to the diffusion of non-charged particles across a membrane, as described by Fick's Laws [19]:

$$J = -D\frac{\partial \varphi}{\partial x} \tag{5}$$

where $J$ is the net flux across the membrane, $D$ is the diffusion coefficient ($D = 1$ here) and $\frac{\partial \phi}{\partial x}$ is the concentration difference across the membrane.

The process reaches equilibrium, indicated by a null net flux of particles across the membrane, when the concentration of particles on both sides becomes equal. In simpler terms, this occurs when the same proportion of samples is found above and below the central value, $A = B = 1/2$. This situation arises when $m_x$ itself equals the true median $m$.

By exploiting this natural dynamic, our algorithm provides an unbiased estimation of the signal's median.

## Hardware

In Fig 2 we presented the hardware implementation of our algorithm. Compared to the state-of-the-art, area-efficient 1-D median filter [13–16, 20], our design features a simpler and smaller cell. More importantly, our algorithm estimates the median with the same estimator variance as state-of-the-art estimators, but it requires a buffer that is significantly smaller (approximately eight times less). This leads to reduced hardware area and power consumption.

In Table 1 we present a comparison with previous work.

There are two classes of median filters: word-level and bit-level. In the word-level approach, each sample is compared to every other sample in the buffer [13–17]. In contrast, the bit-level scheme employs a cascade of partial units that compare or select from the last L samples, starting with the most significant bit of each sample. This process continues, narrowing the scope of estimation until the correct median sample is identified. The bit-level scheme does not require maintaining a sorted buffer [20–23], resulting in better area efficiency and lower

**Table 1. Hardware comparison.**

| Architecture | # clocks per word | Hardware Complexity | Latency | Frequency(L) | $\sigma^2/L$ (norm) |
|---|---|---|---|---|---|
| Moshnyaga [13] | 2 | $\mathcal{O}(L)$ | 1 | decrease | 1 |
| Chen [14] | 1 | $\mathcal{O}(L)$ | 1 | decrease | 1 |
| Chen [15] | 2 | $\mathcal{O}(L)$ | 1 | decrease | 1 |
| Nikahd [16] | 1 | $\mathcal{O}(L)$ | 1 | decrease | 1 |
| Lin [20] | 1 | $\mathcal{O}(L)$ | 1 | same | 1 |
| This work | 1 | $\mathcal{O}(L)$ | 1 | same | 8 |

power consumption compared to the word-level median filter. Either storing a sorted buffer or performing bit operations on the last $L$ elements, the area is always proportional to the window size or the number of sample bits (typically 8-bit or 16-bit). Power consumption is proportional to the square of the frequency and the number of clocking transistors, which are also proportional to the buffer length. All the referenced works, whether utilizing word-level or bit-level architectures, rely on the classical moving window median. In contrast, our proposed algorithm achieves the same variance estimator performance with a buffer that is eight times smaller, significantly reducing power consumption and area, thereby outperforming any previously published work.

## Discussion

Quantile estimation is a powerful technique for characterizing the properties of datasets. It allows for the online estimation of various order statistics, including the minimum, maximum, median, quartiles, and any other quantiles (q-quantile). A significant amount of research on computing median estimators in data streams has emerged in the last decade, particularly for application in filtering techniques. These techniques often involve calculating the exact median of a window containing L samples and then using this value to estimate an appropriate threshold. In sequential streaming algorithms, finding the minimum and maximum values requires only $\mathcal{O}(1)$ operations, whereas estimating the median requires at least $\mathcal{O}(\log L)$ operations [24, 25]. Efficient online computation of quantile statistics is mainly based on the work of Greenwald and Khanna [26, 27]. These algorithms are designed to maintain statistics of arbitrary quantiles. While they are efficient in terms of memory usage and exhibit optimal complexity with operations of $\mathcal{O}(1)$, these operations are not well-suited for implementation on field-programmable gate arrays (FPGAs) or application-specific integrated circuits (ASICs). As a result, the complexity and computational cost of implementing these algorithms outside of a traditional computer or embedded system are significantly high.

Although our proposed method is limited to estimating the median, its implementation on FPGAs is remarkably straightforward. This efficiency arises from the fact that it only requires basic operations such as comparisons and shifting, which are highly optimized for FPGAs and GPUs. Furthermore, since parallelization is the primary advantage of FPGA/GPU architectures, comparing all elements in the buffer can be achieved in $\mathcal{O}(1)$ time. Additionally, shifting elements within the buffer also operates in $\mathcal{O}(1)$ time. Our hardware design does not include any cascaded comparator units or logic gates, which has posed challenges in increasing buffer length in some previous works [13, 14]. Our design scales linearly with buffer length in terms of area, while maintaining output latency, similar to the latest state-of-the-art median filter [16]. Our proposed method allows the computation of quartiles using three implementations of the same algorithm. The primary implementation calculates the median, while the two secondary implementations receive only samples that are either greater than or less than the

median, respectively. This approach allows for distribution segmentation at any desired power-of-two level.

## Conclusion

We presented a novel method for estimating the median of a data stream that: (1) for a given buffer length reduces estimator variance by up to eight times compared to classical moving median estimator, (2) for a given estimator variance it needs up to eight times smaller buffer length, which leads to a footprint area that is also reduced by more than eight times, resulting in lower power consumption, and (3) provides a faster reaction to sudden changes in signal distribution.

## Supporting information

**S1 Fig. Sudden change from skewed distributions.** Step median change, from a Beta(1.5, 4) distribution, which has a median of 0.2439 and a mean of 0.2727 (skewed), to a Beta(4, 1.5) distribution, with a median of 0.7561 and a mean of 0.7273. In response to this significant median change, the new median experiences an intrinsic delay of L/2 before it begins to displace older values from the central position of the buffer. In contrast, the CMM exhibits a gradual change characteristic of a sliding window median. Nevertheless, both NM-511 and CMM-511 converge towards the new median value in approximately *L* time steps (see supplementary S3 Fig). However, NM-63 achieves the same variance estimator as CMM-511, with a significantly faster settling time.
(TIF)

**S2 Fig. Estimator robustness against extreme outliers.** Samples were drawn from a Gaussian mixture that includes both the primary distribution and an outlier distribution (noise). Both estimators demonstrate robustness against extreme outliers. The NM-63 outperforms the CMM-63, exhibiting an estimator variance that is eight times smaller.
(TIF)

**S3 Fig. Settling time for changes in the median of the Folded Normal distribution.** A step change was generated in the median, increasing from 0.67 to 1.21, 1.81, 3.01, 4.82, 7.84, 12.66, and 20.50. These medians correspond to the standard deviation of Gaussian noise, as described by Eq (2). We selected NM-63 and CMM-511 because they exhibit similar estimator variance. While CMM demonstrates a consistent settling time across all levels of step change, which is closer to its buffer length of 511, the settling time of NM is more influenced by the magnitude of the step change. Nevertheless, NM outperforms CMM at all levels, and with moderate step changes, it begins to approach the settling time of L steps. For each configuration, there were 3,000 iterations. The settling time and standard deviation were defined as explained for Fig 4c.
(TIF)

**S1 Table. Comparison of estimator performance across various input distributions.** The standard deviation of the estimator for various input distributions was analyzed. Each type of distribution was tested with different parameters, revealing that the relationship between the estimator's standard deviation, denoted as $\hat{\sigma}$, and the distribution's standard deviation, $\sigma$, remains consistent across all distribution types and is solely dependent on the buffer length. We tested NM and CMM with buffer lengths of 63 and 511 to demonstrate that NM-63 performs equivalently to CMM-511 across all distribution types.
(TIF)

## Author Contributions

**Conceptualization:** Ariel Burman, Sergio E. Lew.

**Formal analysis:** Ariel Burman, Sergio E. Lew.

**Funding acquisition:** Sergio E. Lew.

**Investigation:** Ariel Burman, Sergio E. Lew.

**Methodology:** Ariel Burman, Sergio E. Lew.

**Project administration:** Sergio E. Lew.

**Resources:** Sergio E. Lew.

**Supervision:** Sergio E. Lew.

**Validation:** Jordi Solé-Casals.

**Writing – original draft:** Ariel Burman, Sergio E. Lew.

**Writing – review & editing:** Ariel Burman, Jordi Solé-Casals, Sergio E. Lew.

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
