## [Decision Letter · Decision Letter 0]

21 Aug 2024

PONE-D-24-29737Robust and Memoryless Median Estimation for Real-Time Spike DetectionPLOS ONE

Dear Dr. Lew,

Thank you for submitting your manuscript to PLOS ONE. After careful consideration, we feel that it has merit but does not fully meet PLOS ONE’s publication criteria as it currently stands. Therefore, we invite you to submit a revised version of the manuscript that addresses the points raised during the review process.

We look forward to receiving your revised manuscript.

Kind regards,

Palanisamy R

Academic Editor

PLOS ONE

Journal Requirements:

UBACYT 20020220200032BA

AMOUNT USD 320

5. We note that your Data Availability Statement is currently as follows: All relevant data are within the manuscript and its Supporting Information files.

**Additional Editor Comments:**

Dear Author,

Based on the reviewer comments, the manuscript needs Major Revision. Address all the points with step by step responses to the reviewer comments.

Thank you

Reviewers' comments:

Reviewer's Responses to Questions

**Comments to the Author**

1. Is the manuscript technically sound, and do the data support the conclusions?

Reviewer #1: Yes

Reviewer #2: Partly

Reviewer #3: No

2. Has the statistical analysis been performed appropriately and rigorously? 

Reviewer #1: No

Reviewer #2: N/A

Reviewer #3: No

3. Have the authors made all data underlying the findings in their manuscript fully available?

Reviewer #1: Yes

Reviewer #2: No

Reviewer #3: No

4. Is the manuscript presented in an intelligible fashion and written in standard English?

Reviewer #1: Yes

Reviewer #2: No

Reviewer #3: No

5. Review Comments to the Author

**Reviewer #1:** need some modifications

1. Need more analysis

2. Literature survey is very less in the paper

3. At least 20 latest references need to add in the paper

4. Highlights should be there for main work

5. Comparison with traditional algorithms is impressive so need to design with proper table and diagram.

**Reviewer #2:** 1. A thorough proofreading of the work is suggested.

2. The introduction does not clearly present the main research contribution.

3. The overall organization of the paper is missing in the introduction.

4. While designed for efficiency, the actual computational complexity might still be higher than other methods under certain conditions or for extremely large datasets.

5. FPGA implementation might have limitations related to hardware resources, such as memory and processing power, that could affect performance.

6. The method’s performance and efficiency could degrade with an increasing number of channels or higher sampling rates.

7. The method might not adapt well to highly non-stationary signals or sudden, drastic changes in signal characteristics.

8. There could be latency introduced by the FPGA or other hardware, potentially affecting real-time performance.

9. The accuracy of the median computation might be affected in noisy environments or with very small window sizes.

10. The proposed method might be complex to implement on different types of hardware or in various software environments.

11. Despite eliminating the need to store arrival times, the method might still require significant memory for other aspects of the computation.

12. Although designed to reduce sensitivity to outliers, extreme or unusual outliers might still affect performance.

13. There could be challenges related to synchronization and data handling when implementing parallel processing on FPGAs.

14. The method's ability to meet real-time constraints might be affected by the processing speed and hardware performance.

15. The method might not perform as well with signals that have different characteristics than those it was designed for (e.g., different types of noise or interference).

16. The performance of the threshold calculation might be influenced by the choice of parameters, which could require fine-tuning.

17. The effectiveness of the median estimator might be highly dependent on the chosen window size, which could vary with different applications.

18. Handling and processing large amounts of data in real time might introduce errors or data integrity issues.

19. The method's robustness to various types of noise might need further validation and testing.

20. Integrating the new estimator with existing neural recording systems and data pipelines might be challenging.

21. The method’s performance might be affected by the dynamic range of the signals being processed.

22. The estimator might require calibration for different types of signals or recording setups to ensure optimal performance.

23. Comprehensive validation and testing are needed to ensure the method performs reliably across different scenarios and conditions.

**Reviewer #3:** 1. No novelty of the proposed system.

2. The major contributions of the proposed system is missing

3. No architecture for the proposed system.

4. The simulation setting of the proposed system is missing

5. Attacks results are not mentioned and discussed in the article

6. Conclusion needs more clarity.

6. PLOS authors have the option to publish the peer review history of their article (what does this mean?). If published, this will include your full peer review and any attached files.

Reviewer #1: No

Reviewer #2: **Yes: **Jyotir Moy Chatterjee

Reviewer #3: No

---

## [Author Response · Author response to Decision Letter 0]

9 Oct 2024

October 5, 2024

Dear Dr. Palanisamy,

We have received your decision letter on our manuscript (manuscript PONE-D-24-29737) along with comments from two reviewers. We are thankful for the review process our manuscript has received and appreciate the reviewers effort. We acknowledge the issues identified by reviewers and believe that our revised manuscript fully addresses their concerns. All changes and requested additions have been included in this revision. A detailed point-by-point response to the reviewers’ comments is provided below.

Sincerely,

Sergio E. Lew, PhD

Full Professor-University of Buenos Aires

Instituto de Ingeniería Biomédica, Director

IBYME-CONICET

Buenos Aires, Argentina

General response to all reviewers:

We thank the reviewers for their dedication and effort in pointing out the weak aspects of our paper. All reviewers coincide in that the main contribution is missing or little highlighted. We modified the structure of the introduction to highlight the main contribution of our method:

Efficient moving median estimators keep both an ordered buffer of the last N samples and an index buffer with the arrival time of each sample. In a sequential implementation this solution requires O(log N) clock steps while in parallel implementations, complexity reduces to O(1). However, all these implementations rely on the auxiliary information buffer to track the oldest sample in the main buffer.

Our method not only avoids the use of this auxiliary information buffer, but more importantly, it requires a significantly smaller buffer to achieve a given accuracy (measured as the variance of the estimator), drastically reducing both the hardware area-size and power consumption. The algorithm is unbiased and more robust compared to the traditional moving median one and for sudden shifts in the input signal, it adapts to the new probability distribution significantly faster than the traditional method, mainly due to its reduced buffer length. We add figures to the methods section to illustrate the difference between the classical and our algorithm and to show a hardware implementation.

Reviewer #1: need some modifications

We thank the reviewer for his comments and suggestions.

1. Need more analysis

We include a detailed analysis of the method along with comparison to other state-of-the-art algorithms.

We added an explanation of the algorithm dynamic and compared to classical moving median.

We added a hardware implementation of the algorithm to demonstrate its feasibility and simpler design compared to other works.

We added supplementary figures to show the performance of the algorithm when extreme outliers occur to show the performance of the algorithm with sudden changes of the signal characteristics.

2. Literature survey is very less in the paper

3. At least 20 latest references need to add in the paper

We compared our method with state-of-the-art algorithms and we think now the paper covers these two observations. 

4. Highlights should be there for main work

We have rewritten part of the manuscript to highlight the main contributions. We made emphasis on the major advantages of our algorithm, buffer length reduction leading to less area and power requirements.

5. Comparison with traditional algorithms is impressive so need to design with proper table and diagram.

Thanks to the reviewer for this comment, we added a Table 2 to compare our algorithm against the classical median under different distributions. We also added a Table 1 comparing our hardware design against state of the art hardware implementations of classical moving median.

Reviewer #2: 

We would like to express our gratitude to the reviewer for their insightful comments, which have significantly enhanced the quality of our manuscript.

1. A thorough proofreading of the work is suggested.

It was done.

2. The introduction does not clearly present the main research contribution.

The manuscript has been revised to foreground the principal contributions of our research.We made emphasis on the major advantages of our algorithm, buffer length reduction leading to less area and power requirements.

3. The overall organization of the paper is missing in the introduction.

We added a last paragraph in the introduction to cover this point.

4. While designed for efficiency, the actual computational complexity might still be higher than other methods under certain conditions or for extremely large datasets.

We included an analysis to show that for both sequential and parallel implementations the algorithm has always less complexity than other state-of-the-art algorithms. The algorithm takes one sample at the time, stores it in the buffer (dropping another element) and continues with the next sample. A large dataset presents no new issues because the algorithm is prepared to work with streaming data, when samples are generated in real-time.

5. FPGA implementation might have limitations related to hardware resources, such as memory and processing power, that could affect performance.

We added the schematic hardware implementation to show that each processing cell in the buffer has fewer components than other state-of-the-art algorithms due to the fact that it does not need to track the lifetime of samples in the buffer. We also added Table 1 comparing our work against current state of the art implementation of median filters and added an explanation on the reduced area and power requirement against those work, based on the improved variance of the estimator.

6. The method’s performance and efficiency could degrade with an increasing number of channels or higher sampling rates.

Same as 5 but with the addition that as fewer components and required area-size are needed the algorithm will always perform better than others for the same estimator variance.

7. The method might not adapt well to highly non-stationary signals or sudden, drastic changes in signal characteristics.

We thank the reviewer for this insightful comment. We want to clarify that the median statistics exists only for stationary portions of the signal, that is, those periods when the probability distribution can be defined. However, in real applications the signal median can change due to external factors, for example, a change in the electrode impedance or a sudden change in its position in extracellular recordings. In that case we showed that, for a given estimator variance, our algorithm adapts to the new median 8 times faster, mainly due the the fact that the buffer is 8 times shorter that the needed in traditional algorithms. We added a supplementary figure to show how the method responds to a sudden change in the median of a Beta distribution. We select this distribution to show that the algorithm not only works well for normal distributions.

8. There could be latency introduced by the FPGA or other hardware, potentially affecting real-time performance.

Our hardware design does not include any cascaded comparator unit nor logic gate, which has been an issue to increasing buffer length in some of previous works (Moshnyaga and Hashimoto 2009, Chen et al 2013). Our design scales with buffer length linearly in area, without reducing the latency in the output, similarly to the latest state of the art median filter (Nikahd ad Behnam 2016).

9. The accuracy of the median computation might be affected in noisy environments or with very small window sizes.

We studied this case in a simulation with samples drawn from a gaussian mixture containing the main distribution and an outlier distribution (noise), see Suppl. Figure 2. Again, for the same buffer length, our method is more robust against outliers than traditional algorithms. On the other hand, if we fixed a maximum estimator variance, our method needs a buffer 8 times shorter than the traditional methods, see Figure 4b.

10. The proposed method might be complex to implement on different types of hardware or in various software environments.

Our algorithm is simpler than the state-of-the-art ones due to the fact that it avoids the sample lifetime buffer. We think the implementation complexity on different types of hardware or in various software environments should be the same or even less than with other median estimators.

11. Despite eliminating the need to store arrival times, the method might still require significant memory for other aspects of the computation.

We showed in the hardware implementation, Figure 2, that it is simpler than other state-of-the-art algorithms. It needs less memory and less gates and comparators and in consequence smaller area-size and consumption.

12. Although designed to reduce sensitivity to outliers, extreme or unusual outliers might still affect performance.

Please see response to point 9.

13. There could be challenges related to synchronization and data handling when implementing parallel processing on FPGAs.

Figure 2 shows the hardware implementation and again, compared to other algorithms there is no differences in synchronization and data handling.

14. The method's ability to meet real-time constraints might be affected by the processing speed and hardware performance.

We showed in the hardware implementation, Figure 2, that it is simpler than other state-of-the-art algorithms. It needs less memory and less gates and comparators, processing each new sample in one step, so it should not face more issues than traditional algorithms.

15. The method might not perform as well with signals that have different characteristics than those it was designed for (e.g., different types of noise or interference).

We show the estimator behavior for many buffer lengths and different distributions in Table 2. We also tested the algorithm in changing median scenarios, as the one shown in Suppl. Figure 1.

16. The performance of the threshold calculation might be influenced by the choice of parameters, which could require fine-tuning.

Our algorithm requires only one parameter: the buffer length. This is a significant advantage over traditional algorithms, which typically need additional parameters such as sample lifetime buffer and the number of bits used for lifetime coding. Our simplified design eliminates the complexity associated with these extra parameters.

Concerning the threshold value for spike detection, a SNR = 4 is commonly employed. So, as shown in equation 2, the threshold is a salud version of the median of |x|. 

17. The effectiveness of the median estimator might be highly dependent on the chosen window size, which could vary with different applications.

We show the estimator behavior for many buffer lengths and different distributions in Table 2. 

18. Handling and processing large amounts of data in real time might introduce errors or data integrity issues.

While this affirmation could be true, the same occurs for other median estimators.

19. The method's robustness to various types of noise might need further validation and testing.

We show the estimator behavior for many buffer lengths and different distributions in Table 2. 

20. Integrating the new estimator with existing neural recording systems and data pipelines might be challenging.

Most of the existing neural recording systems have a high pass filter and then a threshold comparator to detect events with higher amplitude (spikes). Our algorithm computes this threshold automatically and could replace the original algorithm with no issues due to it requires fewer resources.

21. The method’s performance might be affected by the dynamic range of the signals being processed.

Our method operates on already sampled data. Saturation is a problem of the amplification system and any other median estimation algorithm would encounter the same issue. There are solutions to detect saturated amplifiers or similar conditions, but this problem is out of the scope of this work, which focuses on presenting the median estimation algorithm.

22. The estimator might require calibration for different types of signals or recording setups to ensure optimal performance.

Our algorithm requires only one parameter: the buffer length. This is a significant advantage over traditional algorithms, which typically need additional parameters such as sample lifetime buffer and the number of bits used for lifetime coding. Our simplified design eliminates the complexity associated with these extra parameters.

23. Comprehensive validation and testing are needed to ensure the method performs reliably across different scenarios and conditions.

We added simulations with different distributions to ensure that the method performs reliably.

Reviewer #3: 

We thank the reviewer for his comments and suggestions.

1. No novelty of the proposed system.

The manuscript has been revised to highlight the principal contributions of our research.

2. The major contributions of the proposed system is missing

Efficient moving median estimators keep both an ordered buffer of the last N samples and an index buffer with the arrival time of each sample. In a sequential implementation this solution requires O(log N) clock steps while in parallel implementations, complexity reduces to O(1). However, all these implementations rely on the auxiliary information buffer to track the oldest sample in the main buffer.

Our method not only avoids the use of this auxiliary information buffer, but more importantly, it requires a significantly smaller buffer to achieve a given accuracy (measured as the variance of the estimator), drastically reducing both the hardware area-size and power consumption. The algorithm is unbiased and more robust compared to the traditional moving median one and for sudden shifts in the input signal, it adapts to the new probability distribution significantly faster than the traditional method, mainly due to its reduced buffer length. We add figures to the methods section to illustrate the difference between the classical and our algorithm and to show a hardware implementation.

3. No architecture for the proposed system.

We added a figure showing the hardware implementation.

4. The simulation setting of the proposed system is missing

We include a figure in methods to clarify this point along with a detailed description of both the classical moving median estimator algorithm and ours.

5. Attacks results are not mentioned and discussed in the article

We are not sure what the reviewer means with “Attacks results” but we believe his concerns are covered in the new version of the manuscript.

6. Conclusion needs more clarity.

We improved the conclusions comparing our results with state-of-the-art implementations.

---

## [Decision Letter · Decision Letter 1]

30 Oct 2024

Robust and Memoryless Median Estimation for Real-Time Spike Detection

PONE-D-24-29737R1

Dear Dr. Lew,

We’re pleased to inform you that your manuscript has been judged scientifically suitable for publication and will be formally accepted for publication once it meets all outstanding technical requirements.

Kind regards,

Palanisamy R

Academic Editor

PLOS ONE

Additional Editor Comments (optional):

Dear Author,

Based on reviewer comments, the manuscript recommended to Accept.

Thank you.

Reviewers' comments:

Reviewer's Responses to Questions

**Comments to the Author**

1. If the authors have adequately addressed your comments raised in a previous round of review and you feel that this manuscript is now acceptable for publication, you may indicate that here to bypass the “Comments to the Author” section, enter your conflict of interest statement in the “Confidential to Editor” section, and submit your "Accept" recommendation.

Reviewer #1: All comments have been addressed

Reviewer #3: All comments have been addressed

2. Is the manuscript technically sound, and do the data support the conclusions?

Reviewer #1: Yes

Reviewer #3: Yes

3. Has the statistical analysis been performed appropriately and rigorously? 

Reviewer #1: Yes

Reviewer #3: Yes

4. Have the authors made all data underlying the findings in their manuscript fully available?

Reviewer #1: Yes

Reviewer #3: No

5. Is the manuscript presented in an intelligible fashion and written in standard English?

Reviewer #1: Yes

Reviewer #3: Yes

6. Review Comments to the Author

Reviewer #1: 1. Very impressive manuscript

2. Analysis is also justifying the concept

3. More calculation may be added for more comparisons

4. Need to increase the latest references such as:

Saikat Gochhait, Yogesh Singh Rathore, Irina Leonova, Mahima Shanker Pandey, Bal Krishna Saraswat, Santosh Kumar Maurya, Hare Ram Singh, Nidhi Bansal, “URL shortener for web consumption: an extensive and impressive security algorithm,” Indonesian Journal of Electrical Engineering and Computer Science-Institute of advanced engineering and science, vol. 35, issue 01, pp. 284-291, doi: 10.11591/ijeecs.v35.i1, 2024

Reviewer #3: The author has addressed all the comments suggested by me. Now, a paper quality also improved a lot. So, I am recommending this paper for publications.

7. PLOS authors have the option to publish the peer review history of their article (what does this mean?). If published, this will include your full peer review and any attached files.

Reviewer #1: **Yes: **Nidhi Bansal

Reviewer #3: No

---

## [Editor Report · Acceptance letter]

14 Nov 2024

PONE-D-24-29737R1 

PLOS ONE

Dear Dr. Lew, 

I'm pleased to inform you that your manuscript has been deemed suitable for publication in PLOS ONE. Congratulations! Your manuscript is now being handed over to our production team.

Kind regards, 

on behalf of

Dr. Palanisamy R 

Academic Editor

PLOS ONE